# Prognosis, Controversies and Assessment of Bone Erosion or Invasion of Oral Squamous Cell Carcinoma

**DOI:** 10.3390/diagnostics15010104

**Published:** 2025-01-04

**Authors:** Ahmed Ata Alfurhud

**Affiliations:** Oral and Maxillofacial Surgery and Diagnostic Sciences Department, College of Dentistry, Jouf University, King Khalid Road, Sakaka 72388, Saudi Arabia; aalfarhood@ju.edu.sa or dr.aalfurhud@jodent.org; Tel.: +966-501738443

**Keywords:** cortical bone, medullary carcinomas, oral squamous cell carcinoma, outcome, prognosis

## Abstract

**Objectives**: To discuss the prognostic outcomes, controversies and assessment of bone erosion or invasion of oral squamous cell carcinoma (OSCC). **Methods:** A structured literature review was conducted to critically analyse relevant evidence. The Web of Science database was searched using specific keywords aligned with the review question. After identifying initial studies, their references were also reviewed to include any additional relevant publications, ensuring a comprehensive evaluation of the available evidence. **Results:** The search identified 11 relevant studies, including 5 from the initial search and 6 from reference review. The significance of bone involvement is unclear in OSCC, with varying definitions of cortical bone erosion and medullary bone infiltration contributing to conflicting results regarding the prognostic significance of bone involvement. The majority of evidence stems from retrospective cohort studies without clear study criteria and a lack of power to draw valid conclusions. **Conclusions:** There are currently a lack of high-quality studies assessing bone invasion in OSCC. While there appears to be some evidence that medullary bone infiltration is prognostic, further well-designed studies are warranted.

## 1. Introduction

Oral squamous cell carcinoma (OSCC) is the most common malignant tumour arising in the oral cavity [1]. It is usually preceded by dysplastic changes in the stratified squamous epithelium lining the mouth [2]. OSCC has a wide range of clinical presentations, which makes the diagnosis difficult in some cases, particularly in the early stages [3]. Tumours may also arise in close proximity to the maxillary and mandibular bone.

According to the Tumour Node Metastasis (TNM) staging classification, OSCC that invades the cancellous bone (medullary bone infiltration) is upgraded to T stage T4a (regardless of size), whereas cortical bone erosion limited to the cortical plate does not result in a T stage upgrade to T4a. If no bone involvement has occurred or there is only superficial cortical bone erosion, the T stage is based on the size and depth of the tumour or invasion into adjacent structures [4].

In the present paper, the term ‘Bone Involvement’ refers specifically to two distinct patterns: (1) cortical bone erosion and (2) medullary bone infiltration. However, distinguishing between these two patterns of bone involvement can be challenging in certain cases, with significant implications for both treatment and prognosis.

There are a number of published studies describing the significance of bone involvement, but unfortunately, the majority of them do not sufficiently distinguish between the two patterns of bone involvement [5], which is important to achieve optimal surgical margins, to determine which patients may require adjunctive therapy and to correctly stage patients [6,7].

Medullary bone infiltration involves complete perforation of the cortical bone and extension into the medullary cavity, and establishing this can be challenging, particularly in the erosive pattern [8], as the bone of the maxilla and the mandible have naturally occurring perforations. In edentulous patients, there is atrophy of the maxillary and mandibular bones due to physiological remodelling which happens after tooth extraction [9]. This can also be problematic because it may be difficult to differentiate between patients with bone involvement and those with bone loss due to periodontal disease and bone resorption following tooth extraction [10]. The overall aim of this study was to review and discuss the prognostic outcomes, controversies and assessment of bone erosion or invasion of OSCC.

## 2. Materials and Methods

This study employed a narrative review design to analyse the existing literature, aiming to identify knowledge gaps and provide a comprehensive overview of the prognostic implications, controversies, and assessment of bone erosion or invasion in OSCC. A systematic search was conducted in the Web of Science and PubMed databases using the keywords: “cortical bone”, “medullary carcinomas”, “oral squamous cell carcinoma”, “outcome”, and “prognosis”, combined with the Boolean operator “AND” to ensure comprehensive retrieval of relevant studies. The references of the identified studies were also manually reviewed to identify any additional relevant research that met the specified selection criteria.

The inclusion and exclusion criteria for references were carefully designed to ensure the selection of relevant studies while aligning with the research objectives. The timeframe for inclusion focused on research articles published between 1990 and 2020, as the research and writing process began after this date. Specifically, studies that addressed the role of bone erosion or invasion as an independent prognostic factor and its association with survival rates and poor clinical outcomes were included. The analysis was further refined to focus on studies providing evidence linking cortical or medullary bone invasion by OSCC to adverse prognostic outcomes. No restrictions were applied to study design, allowing for the inclusion of all studies that discussed the significance of bone erosion and invasion, regardless of methodology. However, commentaries and empirical articles were excluded to maintain consistency in the type of evidence analysed, and unpublished studies were excluded to ensure the reliability of peer-reviewed and publicly accessible work. Although no language restrictions were applied during the search, all identified articles were published in English, and no relevant articles in other languages were found.

## 3. Results

The initial database search identified five relevant articles. An additional six studies meeting the inclusion criteria were identified through a review of the references, resulting in a total of 11 studies included in the analysis. A summary of the selected studies is provided in Table 1.

## 4. Review of Literature on Bone Invasion

The controversies surrounding patterns of bone invasion—specifically, cortical bone erosion and medullary bone infiltration—have been discussed to determine whether bone erosion or infiltration by OSCCs is an independent prognostic factor [19]; however, the studies have shown conflicting results [5]. For instance, it has been argued the survival rate difference between two surgical procedures (marginal and segmental mandibulectomy) give no significant difference in the survival rates between these two procedures, even for cortical bone erosion or medullary bone infiltration [6]. In contrast, a strong relationship between the patterns of bone involvement and poor outcomes has also been reported in the literature, showing a correlation with the extent of bone invasion [5,11,12,13,14]. However, research from other authors does not support this position [6,7,20].

Ebrahimi et al. [5] conducted a retrospective cohort study by dividing the patients based on: 1. absent (without bone involvement); 2. cortical bone erosion (limited to cortical plate); and 3. medullary bone infiltration (extension into cancellous bone), and reported that patients with medullary bone infiltration had significantly lower 5-year overall survival (29% vs. 65%) and disease-specific survival (DSS) (36% vs. 77%) compared to those without bone involvement (*p* < 0.001). In contrast, patients with cortical bone erosion alone showed no significant difference in survival compared to those without bone involvement. Distant metastases were more common in patients with medullary bone infiltration (9.8%) compared to cortical bone erosion (4.9%) and no bone involvement (3.8%). Univariate analysis showed a 330% higher risk of distant metastases in patients with medullary bone infiltration versus those without bone involvement. Therefore, medullary bone infiltration was an independent indicator of reduced overall and DSS and had a worse prognosis compared with the others.

Fried et al. [11] investigated the impact of bone involvement in small OSCCs using histopathological criteria to evaluate the depth and severity of invasion. Patients were divided into three cohorts (details in Table 1). Cohorts 1 and 2 demonstrated similar clinical outcomes regardless of bone involvement, whereas cohort 3 exhibited worse overall survival, DSS, distant metastasis and regional control. The study concluded that small tumours with cortical bone erosion may have outcomes comparable to T1 and T2 tumours, but medullary bone infiltration is a significant prognostic factor associated with poorer control and survival rates. Medullary bone infiltration was identified as the strongest predictor of poor outcomes.

Another retrospective study was conducted to assess bone involvement. The results showed a 5-year observed survival rate of 60.35%. Local recurrence (LR) was significantly lower in early-stage tumours (*p* = 0.02). Patients undergoing bone resections greater than 4 cm had lower survival rates compared to those with resections of less than 4 cm (*p* = 0.01). Advanced-stage tumours (*p* = 0.006) and involvement of surgical margins (*p* = 0.0001) or bone (*p* = 0.003) were also significantly associated with reduced survival [12].

Jones et al. [13] found that patients without bone involvement had a 5-year survival rate of 53%, while those with suspected bone involvement had a significantly lower rate of 25% (*p* < 0.02). Bone involvement is a strong indicator of poor prognosis in oral cavity cancers. The findings of Shaw et al. [14] highlight the critical role of bone involvement in predicting recurrence and DSS. The 5-year DSS was 68%, and the crude survival rate was 50%. Notably, the pattern of bone involvement proved to be a significant prognostic factor. Tumours with cortical bone erosion were associated with a relatively favourable prognosis, while medullary bone infiltration indicated more aggressive tumour behaviour and worse outcomes.

However, contrary to previous findings, various theories in the literature suggest differing perspectives on bone involvement. Patel et al. [6] demonstrated that the presence of medullary bone infiltration and the extent of bone invasion did not significantly affect the 5-year local control rate. Similarly, no differences in local control or survival rates were observed between cases with cortical bone erosion versus medullary bone infiltration. However, LR was found to be associated with soft tissue margin involvement in the context of bone involvement. The study concluded that soft tissue margin involvement, when coupled with bone involvement, is the most critical indicator of poor prognosis.

This view is supported by Chen et al. [7], who analysed local control rates in two groups. Among patients with bone involvement, the local control rate was 85.7%, compared to 77.8% in patients without bone involvement. Although the bone involvement was categorized as “None”, “Cortex” and “Medullary” (the latter represented by only one patient), no statistically significant difference was observed between the groups.

Mücke et al. [15] reported findings consistent with those of Patel et al. [6] and Chen et al. [7], showing that bone involvement was not an independent predictor of survival. The mean survival for patients with bone involvement was 71.6 ± 46.2 months, compared to 72.9 ± 48.1 months for those without bone involvement. They concluded that overall survival was primarily influenced by factors such as the extent of mandibulectomy, age, tumour stage, N stage, recurrence and tumour grade, while bone involvement had little impact on prognosis. Thus, although bone involvement does not directly affect survival, the extent of bone involvement in mandibular tumours remains crucial for surgical planning and prognosis. The study of Mücke et al. [15] highlights that while bone involvement is not significantly correlated with survival in OSCC patients, achieving tumour-free margins through surgery tailored to the tumour’s extent is essential for better outcomes.

Three studies have presented conflicting results regarding the impact of bone involvement in OSCCs. Du et al. [16] reported a significantly higher risk of recurrence in OSCCs with bone involvement, although DSS rates were similar regardless of bone involvement. Fives et al. [17] found that medullary bone infiltration significantly worsened local control and overall survival in tumours ≤ 4 cm. In contrast, Petrovic et al. [18] concluded that bone involvement did not adversely affect prognosis, as local recurrence-free survival and DSS were comparable between OSCCs with and without bone involvement.

## 5. Discussion

The study by Ebrahimi et al. [5] underscores the critical prognostic role of bone involvement in patient outcomes. Medullary bone infiltration, as identified in their cohort, was associated with markedly reduced overall and DSS, highlighting its aggressive clinical nature. Moreover, the significantly increased risk of distant metastases, with a 330% higher likelihood compared to patients without bone involvement, further establishes medullary infiltration as a key indicator of poor prognosis. These findings suggest that medullary bone infiltration not only serves as an independent prognostic factor but also emphasizes the need for more aggressive therapeutic strategies for affected patients. In contrast, the finding that survival rates were similar between patients with cortical bone erosion and those without bone involvement suggests that cortical erosion has a less significant impact on prognosis compared to medullary bone infiltration. This highlights that different types of bone involvement affect patient outcomes to varying degrees. These distinctions provide valuable insight into tailoring treatment strategies and improving risk stratification for patients with varying degrees of bone involvement.

Based on the findings of Ebrahimi et al. [5], it is recommended that the T staging system be revised to enable better stratification for DSS. In their study, an alternative T staging system was proposed to more accurately reflect the prognostic differences associated with varying degrees of bone involvement. This system initially classifies tumours based on size as T1, T2, and T3, with the recommendation that tumours should be upstaged by one category (e.g., from T2 to T3) in the presence of medullary bone infiltration. This recommendation is based on evidence showing a significant increase in mortality associated with medullary bone infiltration, regardless of the tumour’s size. Importantly, when bone involvement was only categorized as “present” or “absent”, it was not identified as a significant independent predictor of overall survival or DSS. In contrast, medullary bone infiltration was shown to be a critical criterion for tumour staging.

The current American Joint Committee on Cancer (AJCC) T staging system does not adequately stratify DSS outcomes, as there is considerable overlap between T3 and T4 tumours. However, the alternative T staging system proposed by Ebrahimi et al. [5] provides better stratification of patient prognoses. This highlights the need to establish standardised criteria that differentiate patterns of bone involvement, ensuring that not all tumours with bone involvement are automatically classified as T4.

The findings of Fried et al. [11] align closely with those reported by Ebrahimi et al. [5], further emphasising the critical prognostic implications of medullary bone infiltration OSCC. Both studies highlight that while cortical bone erosion may not significantly affect survival outcomes, medullary involvement is strongly associated with poorer prognosis, including reduced survival rates and increased distant metastasis. Fried et al. [11] emphasised that small tumours (<4 cm) with bone involvement, currently classified as T4 under the AJCC staging system, exhibit outcomes comparable to small tumours without bone involvement. These findings reinforce the need to differentiate between cortical bone erosion and medullary bone infiltration within the staging framework. Furthermore, Fried et al. [11] validated a modified T staging system using their dataset, which offered improved stratification of patient outcomes compared to the existing AJCC system. Their study supports the consideration of modifications to the AJCC staging system to better reflect the prognostic differences between types of bone involvement, thus enhancing the accuracy of staging and treatment planning.

Muñoz Guerra et al. [12] investigated bone involvement in OCSCC using preoperative clinical exams, panoramic radiographs, and/or CT scans, with confirmation based on the extent of postoperative bone resection. However, their study lacked differentiation between cortical bone erosion and medullary bone infiltration and did not specify histopathological criteria for evaluating bone involvement. Compared to the more rigorous methodologies employed by Ebrahimi et al. [5] and Fried et al. [11], particularly in histopathological assessment, the approach used by Muñoz Guerra et al. [12]. was less reliable. This discrepancy in assessment methods raises the possibility that bone involvement may have been underestimated in the study by Muñoz Guerra et al. [12]. The findings highlight the importance of employing standardised criteria to define cortical bone erosion and medullary bone infiltration in future studies. Such standardization would not only improve the accuracy of bone involvement assessments but also enhance the comparability of findings across studies. Additionally, it would contribute to the establishment of more consistent prognostic and staging frameworks for OCSCC, ultimately aiding in the management and treatment of this disease.

Intra and inter-rater reliability is a known issue in pathology where scoring of dysplasia and grading of tumours can give varying results. For example, in numerous fields of pathology, particularly breast and prostate grading, different experienced pathologists favour different features when grading, leading to poor reproducibility [21,22]. It is not unreasonable to assume that similar issues exist when pathologists assess tumour for bone involvement, although no studies examining this issue were identified.

The methodology used in the study by Jones et al. [13] relied mainly on clinical and radiographic examinations to assess bone involvement, without providing a detailed description of the bone involvement patterns (e.g., erosion versus true infiltration). Instead, bone involvement was simply classified as either invaded or non-invaded. While the authors considered clinical examination a safe and reliable method for detecting bone involvement, this approach has limitations. Radiological and clinical assessments may fail to identify smaller or subtler areas of bone involvement that could be detected through microscopic examination. Histopathological analysis, as a more sensitive and precise method, allows for a clearer identification of invasion patterns and could detect smaller areas of invasion that radiological and clinical investigations could miss. This highlights the need for incorporating microscopic evidence into assessment protocols to improve diagnostic accuracy and ensure comprehensive evaluation of invasion patterns. Accordingly, it is suggested that the pattern of invasion was not classified as cortical bone erosion or medullary bone infiltration because the authors focused more on the clinical and radiographic examination without considering that histopathological examination can be better utilized to differentiate between the two patterns.

Some studies have been unable to demonstrate the prognostic significance of bone involvement, as discussed earlier. However, in the study by Patel et al. [6], the authors did not provide clear definitions or criteria for distinguishing between cortical bone erosion and medullary bone infiltration. Additionally, they acknowledged the absence of histological data regarding medullary bone infiltration. In some cases, the determination of bone involvement may rely on discussions between pathologists and radiologists, particularly in cases where the diagnosis is unclear or difficult to assess.

The pattern of bone involvement was assessed using the same criteria in two studies [7,14], yet their results were conflicting. According to Chen et al. [7], histological evidence of bone involvement did not significantly affect local tumour control or overall survival. However, the sample size in this study was relatively small compared to other studies, and the presence of medullary bone infiltration in only a single patient was insufficient to reliably determine its prognostic impact. To draw valid conclusions or demonstrate meaningful differences between cohorts, studies should have adequate statistical power. Without this, the findings may not be reliable due to the study being underpowered. Furthermore, if the criteria used by Ebrahimi et al. [5] and Fried et al. [11] were applied to the cohort in Chen et al. [7], where bone involvement was almost absent, the findings would likely align with those of Ebrahimi et al. [5] and Fried et al. [11]. This is because the presence of medullary bone infiltration is a critical factor in determining the prognostic significance of bone invasion.

The study by Mücke et al. [15] concluded that bone involvement should not be considered an independent predictor of survival. However, their assessment of bone involvement was based solely on whether the bone was invaded or not, without considering the specific pattern of bone involvement. Additionally, bone involvement was evaluated using a combination of clinical examination, CT and MRI, but histopathological examination was not included. As a result, true bone involvement may not have been accurately assessed, which could explain why the study did not find a significant impact of bone involvement on survival.

A systematic review with meta-analysis conducted by Li et al. [19] provided significant insights into the differences between cortical bone erosion and bone infiltration in OSCC. The study concluded that medullary bone infiltration is distinct from cortical bone erosion, with medullary bone infiltration associated with a higher rate of distant metastasis. This increased metastatic potential is attributed to the tumour’s ability to access the circulation through the blood vessels in the cancellous bone. The review included 18 studies with a total of 3756 participants. Among these, 7 studies were classified as having an unclear risk of bias, while the remainder were deemed to have a high risk of bias. The meta-analysis revealed a significant relationship between medullary bone infiltration and overall survival (*p* = 0.04). Notably, medullary bone infiltration was found to significantly decrease overall survival (*p* = 0.0001), while cortical bone erosion showed no significant effect (*p* = 0.66). When focusing on DSS, mandibular medullary bone infiltration involvement was identified as a predictor of poor DSS (*p* < 0.0001), whereas cortical bone erosion did not impact DSS (*p* = 0.66). These findings underscore the importance of differentiating between the two types of bone involvement in prognostic assessments for OSCC.

In the paper by Jimi et al. [23], the authors describe the “the erosive pattern of bone invasion as is marked by a broad pushing front, a sharp interface between tumour and bone, osteoclastic bone resorption and fibrosis along the tumour front and an absence of bone islands within the tumour mass”. However, this definition is problematic. Bone involvement can also present with a “pushing front” and “sharp interface” when a cohesive tumour pushes through the cortical bone into the medullary bone. As such, labelling this as an “erosion pattern” is misleading, as it may suggest that bone involvement is only associated with a discohesive invasive front. This could lead other studies to incorrectly conclude that bone involvement can only occur in such a manner, when in fact it may occur with different patterns of invasion.

In reviewing the literature, the included studies showed conflicting results, which may be due to differences in either the cohort or the methodology of assessment. One major factor is the variation in cohort characteristics, such as the distribution of tumour sites. For instance, in the studies by Ebrahimi et al. [5] and Fried et al. [11], tongue cancer was reported in 204 and 90 patients, respectively, whereas in the cohort of Patel et al. [6], only 9 cases of tongue cancer were included. Furthermore, the number of cases with involved margins was much higher in the studies by Ebrahimi et al. [5] and Fried et al. [11], with 79 and 63 cases, respectively, compared to just 16 cases in Patel et al. [6]. In contrast, the study by Chen et al. [7] did not include any cases involving the tongue or floor of mouth. These differences emphasize the importance of ensuring that patients’ clinicopathological and demographic characteristics are comparable in order to draw reliable conclusions and minimize confounding factors, both within individual studies and when comparing studies. In addition to differences in patient characteristics, the cohort size was much smaller in some studies compared to others. This difference in cohort size could be a contributing factor to the conflicting results observed across these studies.

For future research, developing an objective set of criteria to differentiate between cortical bone erosion and medullary bone infiltration is essential for advancing future research and improving clinical diagnostics. Key areas of focus include analysing osteoclastic activity through immunohistochemical markers to assess the density and presence of osteoclasts in affected regions. Evaluating bone-lining cell integrity using markers can help distinguish reactive processes associated with erosion from infiltrative processes indicative of invasion. Additionally, studying the peritumoural fibro-osseous response may reveal that erosion often involves minimal desmoplasia, while medullary bone infiltration is typically associated with a pronounced desmoplastic reaction. The tumour-bone interface may also be examined, with erosion showing smooth borders and non-infiltrative morphology, while medullary bone infiltration presents irregular, spiculated borders and infiltrative tumour clusters. Advanced imaging techniques, such as micro-CT or confocal microscopy, can further enhance understanding by providing three-dimensional visualization of the bone-soft tissue interface. Future efforts should aim to validate these criteria through large-scale studies, ensure consistency in definitions and incorporate quantitative tools for more precise differentiation.

## 6. Conclusions

This review aimed to evaluate the significance of bone involvement in OSCCs and its influence on survival rates, specifically in relation to two distinct patterns of bone involvement. Additionally, it sought to examine whether consistent definitions exist for cortical bone erosion and medullary bone infiltration. The findings highlight that it is currently not possible to confirm whether bone involvement is a definitive prognostic indicator due to conflicting results and inconsistencies in the definitions of these patterns. The lack of consensus among researchers and pathologists on histological criteria for assessing bone involvement and differentiating between the patterns undermines the reliability of the existing evidence. This ambiguity limits the ability to draw actionable conclusions and implement effective clinical protocols. Despite the previous systematic review conducted in 2017 [19], there remains a critical gap in high-quality evidence to support standardised recommendations.

The practical impact of this review lies in its ability to highlight the challenges in diagnosing and prognosing OSCC patients, while underscoring the need to address current gaps in standardized definitions and assessment criteria. By addressing the lack of standardized definitions for cortical bone erosion and medullary bone infiltration, clinicians can achieve more consistent and accurate assessments of bone involvement, leading to better staging and treatment planning. Clarifying the prognostic significance of these patterns through high-quality studies could enable personalised treatment approaches and more reliable outcome predictions. Additionally, standardised criteria would enhance collaboration among multidisciplinary teams and form the basis for evidence-based clinical guidelines, ultimately ensuring more effective and uniform care for OSCC patients.

To address this, future research must prioritize the development and adoption of standardised definitions and assessment criteria for bone involvement. Such standardization will enable the design of robust, high-quality studies that can clarify whether and how prognosis is impacted by the two patterns of bone involvement.

## Figures and Tables

**Table 1 diagnostics-15-00104-t001:** A summary of studies included in this review.

Reference (Type of Study)	Location of Tumour	No. of Patients	Average Age (Years)	Bone Invasion Assessment
[5] (Retrospective cohort study)	Oral tongue, FOM, Alveolus RMT, Buccal, Hard palate, Other.	498	63.5	The authors clearly defined and assessed the patterns of bone involvement, categorizing it into three distinct types: (1) No Bone Involvement (Absent): Absence of any bone involvement. (2) Cortical Bone Invasion: Involvement limited to the cortical layer of the bone. (3) Medullary Bone Invasion: Extension into the cancellous (spongy) bone.
[11] (Retrospective cohort study)	Alveolar ridge, Buccal, FOM, Gingiva, Hard Palate, Oral tongue, RMT	254	---------	Patient Cohorts Based on Tumour Size and Bone Invasion: Patients were classified into three groups according to primary tumour size and bone invasion status: (1) ≤4 cm Without Bone Invasion: Tumours ≤ 4 cm in size with no evidence of bone involvement (AJCC T1/T2). (2) ≤4 cm With Bone Invasion Only: Tumours ≤ 4 cm in size where bone invasion was the sole factor contributing to an AJCC T4 classification. (3) >4 cm or Additional T4 Factors: Tumours > 4 cm in size or those with other features (e.g., skin invasion or deep muscle invasion) qualifying for an AJCC T4 classification, regardless of bone invasion pattern.
[12] (Retrospective cohort study)	FOM, Gingiva, RMT, Tongue, Buccal mucosa, others	106	57.7	In this study, bone involvement was evaluated preoperatively using clinical examination, panoramic radiographs and/or computed tomography scans. Postoperatively, bone involvement was confirmed based on the extent of bone resection. However, the researchers did not differentiate between the two patterns of bone involvement: cortical erosion or medullary bone infiltration histopathology, and no specific histopathological criteria for assessment were provided.
[13] (Retrospective cohort study)	Base of tongue, Soft palate, RMT, Buccal mucosa, Lateral tongue, Anterior and Lateral FOM, others.	82	Men: 59 Women: 63	The authors assessed bone involvement through clinical and radiographic examinations, with postoperative confirmation based on pathological reports. However, they did not specify whether the bone involvement pattern was cortical or medullary, nor did they provide clear criteria for pathologists to distinguish between these patterns. Data on two indicators of bone involvement were analysed: clinical examination findings, such as tumour fixation to the bone, and radiological evidence of bone involvement.
[14] (Retrospective cohort study)	FOM, RMT Alveolus, Tongue, Buccal or Cheek	100	63	The medullary infiltrative pattern, the tumour penetrates the cancellous bone spaces as small islands or projections, without an intervening layer of connective tissue and with minimal osteoclast activity. In contrast, the cortical erosive pattern features tumour invasion along a broad front, where a connective tissue layer and active osteoclasts separate the tumour from the bone.
[6] (Retrospective cohort study)	-----------	111	63	While the patterns of bone involvement were evaluated, the authors did not specify distinct features for “cortical bone erosion” and “medullary bone infiltration.” According to the authors, the cortical bone erosion was considered to occur when the tumour was clinically observed to be “adherent to” or “superficially involving” the bone. In contrast, the medullary bone infiltration was identified when the tumour extended deeply into the medullary cavity or when the mandible was atrophy.
[7] (Retrospective cohort study)	Buccal mucosa, RMT, Lower gingiva, Lip	43	49.4	The pattern of bone involvement was assessed using the same criteria as described in [14].
[15] (Retrospective cohort study)	Tongue, FOM Alveolar crest, Buccal region	982	60.3	The bone involvement was assessed using the methodology described in [6]. Although the evaluation categorized bone as either invaded or not invaded, the authors did not establish clear histopathological criteria to differentiate between cortical bone erosion and medullary bone infiltration.
[16] (Retrospective cohort study)	Lower gingiva	142	62.7	The assessment of bone involvement was based on a thorough perioperative evaluation, clinical and imaging examinations, intraoperative frozen section analysis, tumour proximity to or fixation on the underlying bone, and the depth of bony invasion. Histopathological analysis confirmed the presence of bone involvement, which the authors categorized into cortical and medullary types. However, they did not provide a clear definition or criteria for distinguishing between these two patterns of bone involvement.
[17] (Retrospective cohort study)	FOM, Lower alveolus, RMT	96	62.6	The extent of bone involvement was categorized as either macroscopic (visible destruction of bone by the tumour observed during gross examination of the specimen) or microscopic (detected only under a microscope). The level of bone involvement was classified into two categories: cortical only (limited to the outer bone layer) and medullary (involving malignant cell infiltration into the cancellous bone of the medullary cavity).
[18] (Retrospective cohort study)	Buccal Mucosa FOM, Lower Gum, RMT, Tongue	326	64	The bone involvement was recorded as cortical and medullary. However, no clear criteria were mentioned, except in medullary bone infiltration when the cancer cells reach into medullary spaces.

Abbreviations: FOM, floor of mouth; RMT, Retromolar trigone; AJCC, American Joint Committee on Cancer.

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
