# Peer review of "Prognosis, Controversies and Assessment of Bone Erosion or Invasion of Oral Squamous Cell Carcinoma"

_diagnostics, 2025, doi:10.3390/diagnostics15010104_

Round 1

Reviewer 1 Report

Comments and Suggestions for Authors

Comments are in the pdf

Author Response

Author's Reply to the Review Report (Reviewer 1):

Comment 1: Keywords must be in alphabetical order.
Response 1: Thank you for pointing this out. I have considered this comment and updated the keywords to: 'cortical bone,' 'medullary carcinomas,' 'oral squamous cell carcinoma,' 'outcome,' and 'prognosis.' I have used MeSH terms as shown in PubMed. The changes can be found on Page 1, point 24.

Comment 2: There is atrophy.
Response 2: Thank you for pointing this out. I have considered this comment and changed “atrophy” to “there is atrophy.” The changes can be found on Page 2, point 49.

Comment 3: This factor is related to the extent of the bone involvement. Thus, if this factor is vital, then even the extent of bone involvement—whether only the cortical plate or medullary bone is involved—will also be critical. Kindly clarify and comment on this aspect here or in the discussion.
Response 3: Thank you for pointing this out. I have considered this comment and added the following clarification:

Mücke et al. (2011) reported findings consistent with Patel et al. (2008) and Chen et al. (2011), showing that bone involvement was not an independent predictor of survival. The mean survival for patients with bone involvement was 71.6 ± 46.2 months compared to 72.9 ± 48.1 months for those without bone involvement. They concluded that overall survival was primarily influenced by factors such as the extent of mandibulectomy, age, tumour stage, N stage, recurrence, and tumour grade, while bone involvement had little impact on prognosis. Thus, although bone involvement does not directly affect survival, the extent of bone involvement in mandibular tumours remains crucial for surgical planning and prognosis. The study by Mücke et al. (2011) highlights that while bone involvement is not significantly correlated with survival in OSCC patients, achieving tumour-free margins through surgery tailored to the tumour’s extent is essential for better outcomes.

The changes can be found on Page 5, points 156 to 166.

Comment 4: Elaborate on the merits and demerits of the currently employed system and suggest points for improvement to formulate an objective set of criteria, as it will help future research.
Response 4: Thank you for your valid and insightful comment. I have addressed this in the manuscript by discussing the merits and demerits of the current system. This has been specifically outlined in Table 1 (Pages 3–4). In particular, Table 1 highlights variability in assessing bone involvement across studies, such as those by Ebrahimi et al. (2011) and Fried et al. (2014), which differ significantly from others. Additionally, the manuscript discusses the lack of clear criteria in some studies, many of which relied on clinical and radiographic assessments.

To improve the evaluation of bone involvement in OSCC, I have proposed the need for universally accepted criteria and precise definitions of erosion and invasion. Future research should focus on consistent cohort characteristics (e.g., tumour site distribution), comparable clinicopathological and demographic features, and adequate cohort sizes.

I have also added a new paragraph suggesting an objective set of criteria to aid future research:

For future research, developing an objective set of criteria to differentiate between cortical bone erosion and medullary bone infiltration is essential for advancing future re-search and improving clinical diagnostics. Key areas of focus include analyzing osteo-clastic activity through immunohistochemical markers to assess the density and presence of osteoclasts in affected regions. Evaluating bone-lining cell integrity using markers can help distinguish reactive processes associated with erosion from infiltrative processes indicative of invasion. Additionally, studying the peritumoural fibro-osseous response may reveal that erosion often involves minimal desmoplasia, while medullary bone in-filtration is typically associated with a pronounced desmoplastic reaction. The tumour-bone interface may also be examined, with erosion showing smooth borders and non-infiltrative morphology, while medullary bone infiltration presents irregular, spic-ulated borders and infiltrative tumour clusters. Advanced imaging techniques, such as micro-CT or confocal microscopy, can further enhance understanding by providing three-dimensional visualization of the bone-soft tissue interface. Future efforts should aim to validate these criteria through large-scale studies, ensure consistency in definitions, and incorporate quantitative tools for more precise differentiation.

The changes can be found in the last part of the discussion (Page 9, points 346–361).

Comment 5: Do you mean medullary invasion?
Response 5: Yes, I mean medullary bone invasion. I have clarified this in the text as:

“The meta-analysis revealed a significant relationship between medullary bone invasion and overall survival (P = 0.04).”

The changes can be found on Page 8, point 308.

Reviewer 2 Report

Comments and Suggestions for Authors

Reviewer’s Comments:

Title:

1) I recommend keeping a more serious tone in the title. It is not necessary to state in parenthesis this is a narrative review. A more appropriate title would be: 

Prognosis, controversies and assessment of bone erosion or invasion of oral squamous cell carcinoma

Or perhaps:

Assessment of bone erosion or invasion of oral squamous cell carcinoma

 Abstract

Consider including in the purpose “assessment of bone erosion or invasion” in oral squamous cell carcinoma”.

Change the word “purpose” to “objectives” as stated in the journal’s author guidelines

Keywords

Keywords should be in alphabetical order. Use MESH terms as keywords.

Materials and Methods:

1. Please  give expand on the justification for inclusion and exclusion criteria of references included. Explain if studies with a specific design were included ? Did the author include commentaries or empirical articles?  Did the author include unpublished studies? The authors' choices should be outlined with appropriate reasoning, even in narrative reviews.

2. Were the articles included only in English?

3. Who selected the articles?

4. Please justify the timeframe established of included manuscripts. Is there likely more evidence that has not been included?

5. Did the papers included address relevant questions?

Results:

1. The results section focuses on bone invasion. This is not in accordance with the overall aim of the review which was to discuss the 1) prognostic features of OSCC, 2) bone invasion and 3) erosion.

Discussion:

1. The discussion section is somehow a repetition of the results section.

2. Please indicate how can the information from this review be applied to clinical practice.

Conclusions:

1. Please rephrase the Conclusions section in such a way that will reflect the practical impact of the review

Author Response

Author's Reply to the Review Report (Reviewer 2): The changes made in response to both reviewers' comments are highlighted in red throughout the manuscript.

Comment 1: I recommend keeping a more serious tone in the title. It is not necessary to state in parenthesis this is a narrative review. A more appropriate title would be: “Prognosis, controversies and assessment of bone erosion or invasion of oral squamous cell carcinoma.”

Response 1: Thank you for your insightful suggestion. I have revised the title accordingly to: “Prognosis, Controversies, and Assessment of Bone Erosion or Invasion of Oral Squamous Cell Carcinoma.” The changes are reflected on Page 1, Point 2.

Comment 2: Consider including in the purpose “assessment of bone erosion or invasion” in oral squamous cell carcinoma. Change the word “purpose” to “objectives” as stated in the journal’s author guidelines.

Response 2: I appreciate your observation and have implemented the suggested changes. The revised section now reads: “Objectives: To discuss the prognostic outcomes, controversies, and assessment of bone erosion or invasion of oral squamous cell carcinoma (OSCC).” The updated text can be found on Page 1, Points 11 and 12.

Comment 3: Keywords should be in alphabetical order. Use MeSH terms as keywords.

Response 3: Thank you for this suggestion. I have updated the keywords to ensure alphabetical order and alignment with MeSH terms: ‘cortical bone,’ ‘medullary carcinomas,’ ‘oral squamous cell carcinoma,’ ‘outcome,’ and ‘prognosis.’ The changes are reflected on Page 1, Point 24.

Comment 4: Please expand on the justification for inclusion and exclusion criteria for references. Explain if studies with a specific design were included. Did the author include commentaries or empirical articles? Were unpublished studies included? The authors' choices should be outlined with appropriate reasoning, even in narrative reviews. Additionally: Were the articles included only in English? Who selected the articles? Please justify the timeframe for included manuscripts. Is there likely more evidence that has not been included? Did the papers address relevant questions?

Response 4: Thank you for these important points. I have addressed them in the revised methodology section:

This study employed a narrative review design to analyze the existing literature, aiming to identify knowledge gaps and provide a comprehensive overview of the prognostic implications, controversies, and assessment of bone erosion or invasion in OSCC. A systematic search was conducted in the Web of Science and PubMed databases using the keywords: "cortical bone," "medullary carcinomas," "oral squamous cell carcinoma," "outcome," and "prognosis," combined with the Boolean operator "AND" to ensure comprehensive retrieval of relevant studies. The references of the identified studies were also manually reviewed to identify any additional relevant research that met the specified selection criteria. The inclusion and exclusion criteria for references were carefully designed to ensure the selection of relevant studies while aligning with the research objectives. The timeframe for inclusion focused on research articles published between 1990 and 2020, as the research and writing process began after this date. Specifically, studies that addressed the role of bone erosion or invasion as an independent prognostic factor and its association with survival rates and poor clinical outcomes were included. The analysis was further refined to focus on studies providing evidence linking cortical or medullary bone invasion by OSCC to adverse prognostic outcomes. No restrictions were applied to study design, allowing for the inclusion of all studies that discussed the significance of bone erosion and invasion, regardless of methodology. However, commentaries and empirical articles were excluded to maintain consistency in the type of evidence analyzed, and unpublished studies were excluded to ensure the reliability of peer-reviewed and publicly accessible work. Although no language restrictions were applied during the search, all identified articles were published in English, and no relevant articles in other languages were found.

The updated methodology can be found on Page 2, Points 57 to 79.

Comment 5: The results section focuses on bone invasion. This is not in accordance with the overall aim of the review, which was to discuss: (1) prognostic features of OSCC, (2) bone invasion, and (3) erosion.

Response 5: Thank you for highlighting this discrepancy. I have made the necessary changes to ensure alignment with the review’s aims. The terminology has been standardized throughout the manuscript to use ‘bone involvement’ as a general term, which encompasses two specific patterns: (1) cortical bone erosion and (2) medullary bone infiltration. Furthermore, I clarified that the prognostic features discussed are specifically related to bone involvement in OSCC.

Comment 6: The discussion section is repetitive of the results section.

Response 6: Thank you for this observation. I have minimized the repetition between the results and discussion sections while ensuring that key information remains for context and coherence.

Comment 7: Please indicate how the information from this review can be applied to clinical practice.

Response 7: Thank you for your comment. This review primarily identifies gaps in knowledge and highlights variations in the literature, rather than providing direct clinical applications. However, I have discussed that not all diagnostic tools are equally reliable for assessing bone involvement. For example, clinical and radiographic assessments are less reliable compared to histopathological evaluation. Additionally, I have emphasised the importance of distinguishing between patterns of bone involvement, such as medullary bone infiltration versus erosion, as this distinction has prognostic significance.

Comment 8: Rephrase the Conclusions section to reflect the practical impact of the review.

Response 8: Thank you for this suggestion. I have revised the Conclusions section to emphasize the practical implications:

The updated text is on Page 10, Points 363 to 388.

Round 2

Reviewer 2 Report

Comments and Suggestions for Authors

s

Please include and clarify in the introduction section that the term ‘bone involvement’  in the present manuscript refers to  two specific patterns: (1) cortical bone erosion and (2) medullary bone infiltration.

Author Response

 Comment 1: Please include and clarify in the introduction section that the term ‘bone involvement’  in the present manuscript refers to  two specific patterns: (1) cortical bone erosion and (2) medullary bone infiltration.

Response 1: Thank you for your suggestion. I have revised the introduction section to clarify that the term ‘bone involvement’ in the present manuscript refers to two specific patterns: (1) cortical bone erosion and (2) medullary bone infiltration. The necessary adjustments have been made, and the changes are highlighted in red on Pages 1 and 2, Points 35 to 61.